# Time Domain (TD) Proton NMR Analysis of the Oxidative Safety and Quality of Lipid-Rich Foods

**DOI:** 10.3390/bios12040230

**Published:** 2022-04-09

**Authors:** Tatiana Osheter, Charles Linder, Zeev Wiesman

**Affiliations:** Plant Lipid Biotechnology Lab (PLBL), Department of Biotechnology Engineering, Faculty of Engineering Sciences, Ben Gurion University of the Negev, Beer Sheva 84105, Israel; osheter@post.bgu.ac.il (T.O.); charles.linder@gmail.com (C.L.)

**Keywords:** chemical and morphology, food safety, oxidation, TD NMR sensor, thermal stress

## Abstract

Food safety monitoring is highly important due to the generation of unhealthy components within many food products during harvesting, processing, storage, transportation and cooking. Current technologies for food safety analysis often require sample extraction and the modification of the complex chemical and morphological structures of foods, and are either time consuming, have insufficient component resolution or require costly and complex instrumentation. In addition to the detection of unhealthy chemical toxins and microbes, food safety needs further developments in (a) monitoring the optimal nutritional compositions in many different food categories and (b) minimizing the potential chemical changes of food components into unhealthy products at different stages from food production until digestion. Here, we review an efficient methodology for overcoming the present analytical limitations of monitoring a food’s composition, with an emphasis on oxidized food components, such as polyunsaturated fatty acids, in complex structures, including food emulsions, using compact instruments for simple real-time analysis. An intelligent low-field proton NMR as a time domain (TD) NMR relaxation sensor technology for the monitoring of T_2_ (spin-spin) and T_1_ (spin-lattice) energy relaxation times is reviewed to support decision-making by producers, retailers and consumers in regard to food safety and nutritional value during production, shipping, storage and consumption.

## 1. Introduction

Food safety is a major and persistent threat to the global status of available foods and is increasingly jeopardizing the effectiveness of public health programs. Unsafe and unhealthy foods can cause a variety of acute and chronic health issues ranging from mild to debilitating, or even life-threatening, conditions. In addition to increased morbidity and mortality, unsafe foods have a significant socioeconomic impact through healthcare costs and lost productivity, as well as reduced trade. There is evidence that foodborne diseases also impact nutritional outcomes, such that food safety includes acute and long-term physiological effects. Relevant health-related nutritional outcomes include gut health, nutrient absorption, growth and metabolic and perinatal development. There is a strong relationship between gastrointestinal illness and growth impairment in children; however, the extent of this impact and its underlying mechanisms have not been completely resolved [1].

Oxidation levels within food products are typically measured by means of tedious, costly and time-consuming laboratory methods [2] that require chemical lab facilities which are often labor intensive, as will be extensively discussed in Section 2. An alternative simple approach is the spectroscopic methodology known as low-field nuclear magnetic resonance (LF ^1^H NMR), which has been used to elucidate the structure of organic compounds containing protons and the porous structures of water-containing inorganics and ceramics.

Data collected by NMR measurements can be presented in the form of multidimensional maps in a relatively easy-to-understand format.

NMR data include the ^1^H spin-lattice energy relaxation time (T_1_) and the spin-spin energy relaxation time (T_2_). These relaxation processes occur when the population of ^1^H nuclear spins return to equilibrium after the absorption of the radio frequency energy generated within the NMR instrument, either by spin–lattice interactions (the lattice/matrix is the nucleus environment of neighboring atoms or molecules) or the mechanism of spin–spin interactions. Thus, T_1_ and T_2_ are the time constants associated with the exponential decay of the magnetization vector until it reaches its equilibrium value: T_1_ indicates how fast the magnetization relaxes back along the *z*-axis (called the longitudinal relaxation time), and T_2_ measures how fast the spins exchange energy in the transverse (x-y) plane (called the transverse relaxation time).

The efficient monitoring of food quality and safety during the entire food chain, from harvesting/breeding, transportation, storage, pre-processing and cooking to the final step of digestion, has not yet been achieved and is a particularly important issue in foods that contain carbon–carbon double bonds, such as those in polyunsaturated alkyl chains, which are found in many foods that are susceptible to deterioration by oxidation into unhealthy products. Current methodologies to readily monitor the resulting chemical and physical changes are not online efficient or readily carried out. To overcome these limitations, time domain nuclear magnetic resonance (TD NMR) technologies have been developed for monitoring oxidative changes in foods that are susceptible to oxidation, for example, many different seeds, oils, emulsions, vegetables, fish and meat products [3,4,5].

Here, we review an efficient methodology for overcoming the present analytical limitations associated with monitoring the composition of food, with an emphasis on oxidized food components, such as polyunsaturated fatty acids (PUFAs), in complex structures, including food emulsions, using compact instruments for simple real-time analysis. This is achieved using an intelligent low-field proton NMR as a time domain (TD) NMR relaxation sensor technology to monitor the T_2_ (spin-spin) and T_1_ (spin-lattice) energy relaxation times, allowing for the analysis of food safety and quality to support decision-making by producers, retailers and consumers regarding nutritional value during production, shipping, storage and consumption. Though there are many reports about TD NMR applications, the ability to use the sensor technology described in this review to produce fingerprints based on chemical and morphological assembly for food safety analysis is neither widely known nor available.

## 2. Current Analytical Methods for Determining the Presence of Unhealthy Food Components and Food Safety

Foods have complex structures whose safety and quality is often related to the detection and minimization of unhealthy components, such as metals, pesticide, preservatives and microbes (i.e., bacteria and fungi), that may be found within the food at different stages of harvesting, delivery, production and storage [6,7]. For example, heavy metals accumulate in plant foods and digestion results in their biomagnification in the human body with an increased risk of neurological, kidney and cardiovascular diseases [8]. Chemical pesticides are used in agriculture to protect crops against insects, fungi, weeds and other pests, and to protect public health by controlling the vectors of tropical diseases, such as mosquitos. However, pesticide residues within food products are potentially toxic upon digestion, resulting in adverse health effects that include cancer and toxic detrimental effects on the organs of the reproductive, immune and nervous systems. Before authorization, pesticides should be tested for all possible health effects and the results should be analyzed by experts to assess the risks to humans [9]. Food preservatives can slow decomposition caused by molds, air, bacteria or yeast [10] in order to maintain food quality and control contamination by foodborne illness, such as life-threatening botulism [9,11]. Microbial spoilage is caused by fungi (i.e., molds and yeasts) and bacteria, whose growth spoils food by producing chemicals that change the food’s color, texture and odor, and make it unfit for human consumption [7]. Postharvest technologies address the issues of the handling, transportation and temperature control of crops, wherein efficient postharvest handling is critical for reducing losses of fresh produce and to maintain food safety and quality, nutrient content and higher market prices [12].

Another basic source of unhealthy components in foods is paradoxically the susceptibility of specific healthy natural components within foods that can undergo reactions with oxygen in the air, forming oxidative products [13,14,15,16]. Many foods are especially susceptible to air oxidation under thermal stress conditions without interaction with the other factors affecting food safety described above. Oxidative processes commonly occur with frying and cooking; however, they also take place during the harvesting, production, shipping and storage of foods with polyunsaturated fatty acid components, such as those found in cheese, butter, milk products and highly susceptible oil-rich foods, such as fish oil and extracted seed oils [17,18,19]. The high susceptibility of oils to oxidative conversion reactions that form unhealthy products is due to their multiple double bonds within alkyl chains [6,13,20,21].

Readily monitoring food safety levels with respect to oxidation is generally difficult because of the chemical and physical/morphological complexity of the food’s susceptibility to oxidation [22]. State-of-the-art analytical systems for the chemical analysis of food oxidation generally requires a wet chemical analysis of extracted samples, including the peroxide value (PV), *p*-anisidine value (AV) and total oxidation (TOTOX) value for the determination of oxidized products, and oxidative stability [2,23,24,25,26,27,28,29,30]. Other chemical analysis methodologies, such as high-performance size-exclusion chromatography (HPSEC), high-field (HF) H^1^ and C^13^ NMR, FTIR, gas chromatography and mass spectroscopy (GC-MS), are complex and require large and costly analytical equipment [22,31,32]. For example, using conventional methodologies, fatty acid methyl esters (FAME) derived from triglyceride food fats and vegetable oils are analyzed by extracting and running samples with gas chromatography [33,34]. In another set of studies, the fatty acid profile, triglyceride composition, organic acid content and lipid oxidation in raw and ultra-high temperature (UHT)-treated milk at 0, 30, 60 and 90 days were determined using GC-MS, and the organic acids were determined by HPLC [35,36]. For the measurement of the induction period of oxidation, a Rancimat oxidation analytical system can be used, where the lipid oxidation is characterized by the peroxide value, *p*-anisidine value and conjugated dienes of free fatty acids [37,38]. As described below, low-field (LF) proton (^1^H) NMR technologies offer the opportunity for direct measurements of food samples for safety analysis, unlike the conventional methodologies described above that generally require sample extraction and analysis with large and relatively expensive instrumentation.

## 3. Low-Field Proton NMR (LF ^1^H NMR) Analysis for Food Safety

Most of the aforementioned analytical measurements require sample extraction and in many cases a purification step. Alternatively, LF ^1^H NMR relaxation analysis has considerable potential in analyzing multicomponent food systems within minutes on intact samples, offering a significant opportunity to advance the field of food science and safety with high accuracy and reproducibility [17,19,22,39,40,41]. This intact rapid sample analysis is performed without separation and purification steps [41,42,43], and uses compact and relatively low-cost LF ^1^H NMR instrumentation. Generally, NMR technology can be divided into both high-field (HF) NMR and LF NMR. HF NMR equipment is expensive and uses large magnetic systems with a high field strength (i.e., 400 MHz) produced by superconducting coils cooled with liquid He, which provide the magnetic fields needed for generating chemical shift material spectra. LF ^1^H NMR, which employs small magnets with a low magnetic field (i.e., 20–60 MHz), is compact and easily used directly on intact samples to measure proton spins and matrix interactions [19,39,44]. However, in some applications LF NMR is not considered to be as accurate as HF NMR instrumentation [22]. Nevertheless, as described below, LF NMR is sufficiently accurate for many useful applications in the safety analysis of complex food structures and contaminating agents [10,19,22,45]. A good example is one report that used LF ^1^H NMR as a rapid, sensitive and highly specific tool for the identification of different microbe-associated components in various foods. This could replace conventional microbial analytical techniques, which are time consuming, costly and require specialized equipment. The rapid detection and quantification of bacteria within the food were carried out by using antibody-functionalized polymer-coated magnetic nanoparticles as a proximity biomarker of the bacteria, which accelerated the decay of the LF NMR relaxation signals and allowed for the measurement of T_2_ spin-spin relaxation [46]. However, this methodology requires microparticles with surface antibodies as ligands for the specific monitoring of microbes during the food’s life cycle.

To utilize the benefits of small, compact and readily used LF ^1^H NMR systems for complex food analysis, without the need for costly and large HF ^1^H NMR equipment, in recent decades LF ^1^H NMR has been developed for the rapid analysis of the chemical and physical/morphological structures related to the health and safety of food [18,22,40]. In this paper, we review both our work and research from others in the development of lab-top portable NMR instruments with a low magnetic strength (LF ^1^H NMR) called time-domain (TD) NMR, which can rapidly measure both the chemical and physical structure/composition of intact food samples as a tool for the analysis of food safety and quality. One important aspect of this is the combinations of the chemical components and their relative structural arrangement, which, in many cases, defines the nutritional value, antioxidant activity and susceptibility to oxidation of components such as polyunsaturated fatty acids (PUFAs) [40,41,47,48]. Recently developed data reconstruction algorithms for TD NMR will be reviewed in subsequent sections; these algorithms can be used for characterizing a material’s chemical and morphological/physical structures on a single 2D T_1_–T_2_ graph of spin-lattice (T_1_) vs. spin-spin (T_2_) relaxation time, wherein individual peaks represent different chemical and morphological structures [47]. The analytical methodology is based on lab-top simple TD ^1^H NMR, which can measure different components, including pathogenic microbes [46,47,49,50]. In this paper, the focus is placed on the unhealthy components generated from the air O_2_ oxidation of nutritional food components with an emphasis on oxidized fatty acid components found within many important food products.

## 4. Chemical and Structural Changes during the Thermal Oxidation of Triacylglycerides and Resulting Unhealthy Components

To understand the complexity of the analytical analysis of oxidized foods and the NMR measurements described in subsequent sections, this section describes the chemical and physical/morphological changes that food undergoes upon oxidation [51,52]. The oxidation process of foods with multiple carbon double bonds is summarized in Figure 1. The different reactions are as follows: The allylic hydrogens on fatty acid chains, found in many foods, are highly susceptible to oxygen reactions, with PUFAs (polyunsaturated fatty acids) being the most susceptible to initiating the oxidation process within oils [48]. The initial heating of oils disrupts their non-covalent secondary cross-linked structures, such as the stabilizing hydrogen bonds of the triacylglycerides (TAGs), and their aggregate structure [6,13,53]. This effect can be easily observed in terms of the reduced viscosity of the heated oils [47]. The initial aggregate disruption is an induction period and is followed by allylic hydrogens interacting with oxygen to initiate the oxidation process by forming peroxides as primary oxidation products [54,55,56,57]. Thus, soon after the disruption of the fatty acid aggregate structures, a reaction between oxygen and the conjugated dienes occurs, and the fatty acid chain forms a hydroxylated and/or epoxidized alkyl chain [58,59,60]. At this stage, a cleavage of the oxygenated chain occurs, forming a wide range of aldehyde and ketone products. The tail side of the broken chain contains a portion of low molecular components that are volatilized into the air. The non-volatile aldehydes are accumulated within the oil as MDA (malondialdehyde), HNA (hydroxynonanoic acid) and many other unhealthy derivative components [61,62,63].

The oxidation cascade propagates and more oxygenated chains are formed. At the end phase of oxidation, there is an accumulation of reactive aldehydes, crosslinked polymers and unhealthy acrylamides [22,64,65,66,67,68]. In addition, the thermal oxidation conditions are also known to yield a high level of free fatty acids and unhealthy trans-fatty acids [69].

For the detection of the unhealthy compounds generated in the oxidation reactions shown in Figure 1, it is worth noting that, for food safety analysis, LF NMR is effective in detecting unhealthy oxygenated α, β-unsaturated aldehydes, such as 4-hydroperoxy-, 4,5-epoxy- and 4-hydroxy-2-alkenals, which are generated in the degradation process of food lipids with omega-3 and omega-6 polyunsaturated groups [19]. These oxidized products are considered as both genotoxic and cytotoxic, and are potential causative agents of cancer, atherosclerosis and Parkinson’s and Alzheimer’s diseases [6].

The rate of lipid degradation in food, especially for lipids rich in polyunsaturated alkyls, increases at high temperatures, and the different oxidative compounds formed are a function of the degree of unsaturation of the alkyl chains [13,47,70]. Another important factor is the effect of multiple phases on oxidation, such as in oil and water, which are found in several foods, such as cereals, milk, eggs, meats, vegetables, etc., wherein the exposure of the surface areas of the lipid phase to oxygen influences the oxidation rates [48,71,72,73,74]. When the surface area of the lipid phase is high in relation to the oil volume involved, higher concentrations of oxygenated derivatives are formed than when the exposed surface area is small. This is a significant issue for oil dispersions, such as micelles and emulsions of lipid oil in water mixtures [3,5,47,62].

## 5. TD NMR Sensor for the Compositional, Physicochemical and Textural Analysis of Food

### 5.1. Advanced TD NMR Applications for Food Quality Anaylsis

TD NMR studies take advantage of the differences in molecular mobility between various food components [19,75], as can be measured by the longitudinal (T_1_) and transverse (T_2_) relaxation times of their protons (^1^H), that are usually associated with water and oils [18,22,39,45,49,76,77]. The spectrum separates multiple types of ^1^H protons based on the deconvolution of T_1_ and T_2_ relaxation measurements, thus providing information on the regions with different structural mobilities in the food’s various material domains. This is important for defining the compositional, physicochemical and textural properties of foods, which are related to the conditions of processing, storage and quality control, as well as other industrial factors for controlling food safety [18,19,76].

The largest application of TD NMR in recent decades has been the analysis of fresh food processing and the final products, and their correlation to food safety [18,52,78,79]. As pointed out by Hills [77], most of these new TD NMR applications are based on ^1^H relaxation time and diffusion measurements, which provide a unique window to study the microstructure of food and the molecular dynamics within complex food matrices that can be correlated with their susceptibility to undergoing oxidation and forming unhealthy products. The potential of TD NMR in food quality control and food safety analysis using novel LF ^1^H NMR devices for ex situ and online measurements, e.g., the new ultrafast methods of multidimensional relaxometry and diffusiometry, has been demonstrated in several studies [18,19,39,80]. For example, a rapid and non-invasive TD NMR technology that could assess the quality (i.e., food composition and adulterations) of intact and processed food products, as well as the effects of processing (i.e., drying, cooking, freezing, curing and salting), storage time and storage conditions on the quality of the different food components, was described in [81]. Other experiments have been performed dynamically, i.e., the product is transformed into the final food product while inside the spectrometer and data acquisitions are performed during this transformation [82]. In the last two decades, TD NMR relaxometry and diffusiometry have been widely used to characterize the fat content and water compartmentalization in animal tissues and foods derived from pork, beef, milk, eggs, broilers and aquatic food products, vegetables, grains and cereal products [17,18,19,44]. Confectionary gels and gelatin-based soft candies formulated using different traditional and novel low-calorie sweeteners have also been studied using NMR ^1^H relaxometry [19].

Our group at BGU recently developed an intelligent ^1^H time domain (TD) NMR application for the testing of nutritional safety with respect to the oxidation status of different food products. This required the generation of chemical and morphological food maps, which was achieved by processing ^1^H TD NMR relaxation time signals using a modified primal-dual interior method for convex objectives (PDCO) optimization solver that included optimized *L*1*/L*2 norm regularization parameters for inverse Laplace transformation (ILT) to rapidly generate the 2D T_1_ – T_2_ fingerprint spectra of organic materials; the different spectrum peaks were then assigned their chemistry and morphologies based on the rigidity/mobility of their proton populations [83,84,85,86]. We further dedicated the sensor to follow the changes in T_1_ and T_2_ relaxation times that occurred during oxidation reactions in food products [40,41,47,83,84]. 2D T_1_ – T_2_ graphs have been shown to effectively characterize the chemical and morphological domains of complex materials [40], such as the 2D spectra generated for oils, i.e., butter, rapeseed oil, soybean oil, linseed oil and many others [40,41]. The same oils have also been analyzed in air, under thermal oxidation conditions. These studies have shown how the different degrees of unsaturation of fatty-acid oils affects their chemical and morphological domains, which subsequently influences their oxidative susceptibility [18,40,41]. This TD NMR sensor platform was also used for the generation of 2D T_1_ – T_2_ fingerprinting spectra for very complex solid lignocellulosic biomass [49]. In another study, the versatility of this TD NMR sensor for food safety analysis was also demonstrated for oil-in-water emulsion products by monitoring their oxidative stability. 2D T_1_ – T_2_ fingerprint spectra of a fresh linseed oil body emulsion (control) and a linseed emulsion enriched with fish oil were compared by exposing the emulsions to thermal autoxidation conditions and obtaining the subsequent changes of their chemical and morphological fingerprints [47]. This technology can be used to ascertain formulations with maximum food safety with respect to oxidative stability.

As discussed above, TD NMR provides important information in regard to the supramolecular structure of triglycerides in different oils [52,87]. In effect, it has been shown that by the bi-exponential fitting of T_1_ – T_2_, two components can be differentiated: the fast-relaxing molecular segment attributed to the polar head and the slow-relaxing part of the more mobile non-polar fatty acid tail. In the case of linseed emulsions, the oil fraction is organized in vesicle-like arrangements coated by a layer of amphiphilic phospholipid surfactants and an oleosin protein. The emulsion’s water phase is associated with the oil’s coating layer and forms the oil’s relatively stable supramolecular aggregated vesicle structure (this issue will be further shown and discussed in Section 5.4, in relation to the TD NMR sensor fingerprinting example of emulsions).

Hwang [22] suggested the development of methods that combine the detection of associated multiple oxidation products for the consistent testing of lipid oxidation. In this respect, the ^1^H LF NMR spectroscopy technology described in the present review paper demonstrates a significant potential in readily elucidating the molecular structures and their physical arrangements within different lipid oxidation products. These chemical and physical arrangements within different lipid oxidation products can assist in the determination of products with minimal oxidation or susceptibility to oxidation, and thus the identification of products with optimal food safety.

### 5.2. Demonstration of TD NMR Fingerprinting of Seeds under Oxidative Thermal Stress Conditions for Determining Their Food Safety

In one study, hummus (chickpeas, *Cicer arietinum*) seeds and linseeds—common lipid-rich foods found world-wide—were analyzed using a TD NMR sensor’s chemical and morphological spectral T_1_ – T_2_ fingerprinting (Figure 1). Using a proton segmental motion approach, according to previous reports [87,88], the peaks along the T_1_ – T_2_ diagonal were assigned to the proton populations of the different segments of the fatty acids within the hummus seeds in the following order, from lower T_1_ – T_2_ to higher values: glycerol, double bonds, aliphatic chains and the tail segment of the fatty acid chains of TAGs. The lignocellulosic rigid fibers were assigned to the peaks with the lowest T_2_ (~1 ms) and T_1_ (~100 ms). The peaks in between the diagonal and the rigid lignocellulose fibers were assigned to proteins. The T_1_ – T_2_ relaxation of proton populations associated with fatty acids were assigned to the motion of the four segments along the diagonal line, as described above.

The TD NMR relaxation numerical values of the T_1_ – T_2_ fingerprints of linseed (LS) control (25 °C) seeds were also determined and compared to the fingerprints of linseeds heated for 20 min at 180 °C (Table 1). The T_1_ and T_2_ values in both the LS samples suggested that only minor chemical and morphological changes occurred after seed roasting. The heating energy increased the ^1^H relaxation time of the fatty acid chains due to the breakage of H bonds in between the chains and an increase in proton mobility; however, the chain’s assignment peak order was not changed.

It was observed that relatively minimal changes were obtained between the control and the linseeds heated/roasted at 180 °C for 20 min. The most rigid lignocellulosic fiber components, initially characterized by a T_1_ range from 13 to 115 ms and T_2_ from 0.6 to 0.7 ms, changed after roasting to a T_1_ value ranging from 24 to 28 ms and a T_2_ value ranging from 1.7 to 12.1 ms. The T_1_ – T_2_ of the control LS proteins changed from 51–4.8 ms to 192.7–2.2 ms after roasting. The relaxation of most of the proton populations of glycerol remained stable, with exception of one glycerol peak whose T_1_ – T_2_ increased after roasting to 179–150 ms. This can be explained by some of the glycerol ester bonds being cleaved by the high roasting temperature with the formation of free fatty acids. The relaxation time of the protons associated with the double bonds and aliphatic segments increased in the roasted LS samples, with an increase in T_1_ – T_2_ for the roasted LS in comparison to the control non-heated LS samples.

These results can be explained by the fact that a relatively short heating time, even at a high temperature, evaporates the water content of the seeds but is not enough to induce the chemical and structural changes typical of oxidation, such as H abstraction, which initiates the oxidation process. This effect was previously explained [47] by the fact that the oil surface within the linseeds is coated and encapsulated by both surfactant components and oleosin proteins, which improve the oil’s durability under oxidative stress. This study demonstrated the relative food safety of linseeds and other similar seeds against oxidation into unhealthy side products due to the chemical and physical arrangements in the seeds, as compared to pure linseed oils in a previous study [88], described below in Section 5.3, that showed the oil readily underwent oxidative conversion into unhealthy products.

### 5.3. Demonstration of TD NMR Fingerprinting of Extracted Seed Oils and Their Oxidation

Compared to oils within seeds, pure extracted oils have different morphological arrangements of their components as well as different associated chemicals, such as antioxidants; both factors should have a strong effect on the oil’s thermal oxidation. In other studies, the TD NMR relaxation fingerprinting of PUFA-rich extracted linseed oil clearly demonstrated the segmental motion approach for the assignment of TAGs, as described in the previous sections on the chemical and physical characterization of linseeds. The rate of relaxation of the oil’s different protons was also differentiated in the magnetic field according to segmental rigidity vs. mobility [85,89]. The most rigid segment in the oil’s TAGs was glycerol (Figure 2C). The second most rigid segment were the double bonds [41]. The aliphatic part of the fatty acid chains was relatively mobile and the tail segment was the freest and most mobile. Since the PUFA-rich linseed in the extracted oil (Figure 2D) is well known to be highly susceptible to thermal oxidation, the chemical and structural changes described in Figure 1 above should be reflected in the fingerprints of LS oil (LSO) exposed to adverse thermal stress. Indeed, after 96 h of exposing LSO to 80 °C (Figure 2E), a significant shift of the reconstructed spectral peaks was obtained followed by a bending effect in which T_1_ > T_2_. This bending effect suggested of formation of polymer end products [88].

The different response profiles of thermal-stressed LSO in comparison to the minimal response observed in intact LS seeds is explained by the oil’s self-assembly in an encapsulated form within the seeds as compared to the open ‘house of cards’-like assembly of the extracted oil [86,90]. Thus, the monitoring of oxidative food safety in seeds vs. the extracted oil shows that oil in seeds has a significantly easier to manage oxidative food safety than extracted oils in all stages of production transportation, shelf-life and cooking.

The self-diffusion coefficient (D) of the heated LSO continuously decreased under thermal oxidative stress, suggesting an increase in the viscosity and rigidity of the sample, as could also be seen for brown LSO oxidized after 96 h (Figure 2B) in comparison to control LSO (Figure 2A). The inverse correlation between viscosity and relaxation time is well known and documented [87].

### 5.4. Demonstration of TD NMR Fingerprinting of Milk and Plant Milk Substitutes

The structural organization of food emulsions as oil-in-water vesicle systems is well documented [3,4,47,72,73,74,91]. These emulsion vesicles are described as oil encapsulated within the interior by different interfacial components, such as the polar head groups of amphiphiles, with their hydrophobic tails within the vesicle’s core isolated from the continuous aqueous matrix environment by their head groups.

In another study using the TD NMR relaxation sensor, T_1_ – T_2_ reconstructed spectral fingerprints of cow milk were generated (Figure 3A). A major T_1_ – T_2_ peak was obtained. This peak was assigned to a proton population of the amphiphilic casein and phospholipid as part of the surface of the milk’s oil vesicles. The strong proton signals in the vesicle surface segments are explained by the high concentration of protons from the amphiphilic vesicle components and the water protons closely associated with the vesicle’s surface [47]. A similar T_1_ – T_2_ reconstructed spectral fingerprint pattern is shown in Figure 3B for a lipid emulsion comprising soybean oil and Intralipid, an approved drug product for parenteral nutrition that is used as a source of fats [92]. In both these cases, small peaks were observed along the T_1_ –T_2_ diagonal and were assigned as some non-encapsulated oils in the emulsion vesicles that increased at high temperatures.

T_1_ – T_2_ reconstructed spectral fingerprints of a control based on fresh linseeds as a plant-based milk substitute emulsion (LSE) before and after heating at 55 °C for 96 h are presented in Figure 3C–F [47]. The exposure of two linseed emulsion formulations to thermal stress conditions showed a significant protection from oxidation. Minimal physicochemical and structural changes were obtained during this thermal stimulation study (Figure 3C,D). Microstructure and vesicle size estimation can also be obtained by TD NMR [93,94]. The T_1_ – T_2_ values of the LSE were changed from 2700–1200 ms at the beginning to 1760–620 ms at the end of the study; this was postulated to be due to vesicle fusion, which was confirmed by a confocal microscopy test (Figure 3E,F). A DLS test further confirmed the relatively small change in the vesicle’s size from 800 to 1360 nm [47]. Only about a 10% reduction in the diffusion coefficient was obtained. No significant increase in the signals of the free and un-encapsulated oil was obtained in the spectral T_1_ – T_2_ fingerprinting (Figure 3C,D). It should be noted that PV, PAV and TOTOX confirmed the minimal oxidation of the oil-in-water emulsion formulations. Furthermore, self-diffusion tests of all the emulsion samples also confirmed very minor physical changes in the emulsions’ vesicles. These results are significantly different than those obtained for the oxidation of pure linseed oil, shown above in Figure 2, wherein a high oxidation occurred, as could be seen by the bending of the T_1_ – T_2_ graph. Thus, the difference between an emulsion and an oil in their susceptibility to oxidation demonstrates the importance of the chemical and morphological arrangements of a material’s components with respect to their susceptibility to oxidation, and the efficacy of the TD NMR sensor in characterizing such arrangements in foods. This demonstrates the efficacy of TD NMR to rapidly and efficiently determine the oxidative food safety properties of a given food combination; in this case, foods with oil-in-water emulsions as compared to pure oil without phase encapsulation and protection against oxidation.

## 6. Summary and Conclusions

This review described the efficacy of a lab-top intelligent TD ^1^H NMR relaxation sensor for the real-time analysis of non-modified food samples as a simple methodology for guiding the processing of food with components susceptible to oxidation and their food safety analysis during transportation, storage and cooking in order to minimizing non-nutritional and unhealthy components.

In the food industry, non-nutritional and potentially unhealthy side products are found in many foods, including specific metals, pesticides, preservatives, microbes and oxidative products, which are present in lipid components with carbon double bonds that readily undergo thermal air oxidation into unhealthy products at different stages from farming to consumption. The ^1^H TD NMR sensor technology described in this review demonstrates how oxidative changes in food can be readily monitored to measure the formation of oxidative products, such as aldehydes and polymerized unhealthy end products. The intelligent TD NMR sensor presented here has been shown to efficiently monitor food quality and safety with respect to the oxidized components of well-known health enhancing foods based on lipid oils with polyunsaturated fatty acids (PUFAs) that have a high susceptibility to auto-oxidation. The efficacy of monitoring food safety was also reviewed with respect to the monitoring of oils and oil-in-water emulsions. Both the TD NMR hardware and critical data reconstruction algorithms for intelligent and simple TD NMR relaxation sensors can support decision-making by producers, retailers and consumers in regard to food safety and nutritional value at different stages of production, shipping, storage and consumption.

An important aspect of food safety can be correlated to a food’s morphological and physical arrangements, which can be readily monitored by the described TD ^1^H NMR sensor. For example, the TD ^1^H NMR sensor can monitor and correlate the internal morphological structures of highly oxidizable linseed oil in comparison to the minimal oxidative response observed in linseed oil-in-water emulsions generated by the sonication of linseeds in water. This demonstrates the management differences in oxidative food safety procedures that are needed for pure oils as compared to the more facile procedures used for oil-in-water emulsions during all stages of production transportation, shelf-life and cooking.

Based on the TD ^1^H NMR sensor databases of lipid oxidation, together with the T_1_-T_2_ fingerprints with a self-diffusion signature of already collected and available data, we suggest that the applicability of this TD ^1^H NMR sensor in the food industry should be further developed for food processing with optimal nutritional value and to monitor food safety against unhealthy oxidative products during the different stages of food production until digestion. Future efforts should be directed in part to developing a machine learning model based on the pattern recognition of TD NMR relaxation chemical and morphological fingerprint changes in TAGs during thermal conditions that stimulate oxidation. Such a well-trained artificial intelligent system may rapidly assess the oxidative safety status of food products and should be applied as a system for food safety decision support (FSDS) to producers and consumers.

## Data Availability

Not applicable.

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
