# Peer review of "Time Domain (TD) Proton NMR Analysis of the Oxidative Safety and Quality of Lipid-Rich Foods"

_biosensors, 2022, doi:10.3390/bios12040230_

Round 1
Reviewer 1 Report
Dear Authors,
unfortunately I was disappointed by the review titled “TD Proton NMR Assessment of Lipid-rich Foods Oxidative Safety and Quality”. While the section describing TD NMR and the work of the group is written well, the introduction and food related parts are written rather poorly. The English used needs strong editing (misspellings and word order). It is unclear, at least to me, what the aim of the review is, and it is therefore hard to read. There is also a central theme throughout the review missing. The authors do not use the commonly understood technical terms in food safety (e.g. it is residues not residuals, it is testing and not assessment (which is a risk management term), therefore, the value to the food safety community is questionable. The authors only cite 48 references for a review article. Out of the 48 references 13 references are published by the authors. A quick search in google scholar with the searchwords “TD proton nmr, food, quality” got me over 26,000 hits. It is therefore unclear which databases they used for their literature search, which keywords they used and how they decided which literature to use and cite. Also some sections look like unpublished own material of the authors (4.3 to 4.5). This is highly unusual for a review article, maybe more appropriate for a rapid communication article. This manuscript therefore does not fulfill the requirements of a literature review.
Author Response
Rev 1: (a point-by-point response to the reviewer’s comments)
Extensive editing of English language and style required:
* As requested, we made an extensive editing of the English and style carried out by an English speaker.
Modification of the title of the review:
*As requested, we changed the title to: " Time Domain (TD) Proton NMR Analysis of Lipid-rich Food’s Oxidative Safety and Quality".
the introduction and food related parts are written rather poorly:
*As requested, we added to the manuscript an introduction section including the importance of food safety for nutrition and health and improved the presentation of the food related parts.
what the aim of the review is, and it is therefore hard to read:
*As requested, the aim of the review was clearly described in the last paragraph of the Introduction section.
The authors do not use the commonly understood technical terms in food safety (e.g. it is residues not residuals, it is testing and not assessment:
*As requested, we change all over the manuscript the technical terms from residual to residues and from assessment to testing, Etc.
The authors only cite 48 references for a review article. Out of the 48 references 13 references are published by the authors:
*As suggested, we added additional 40 relevant references, so our unique research group contribution (13 references) is proportional to the total of 94 references. Furthermore, our previous published papers are essential for demonstration of the unique chemical and morphological fingerprinting spectra of food products that are highly important for analysis of food safety and quality.
some sections look like unpublished own material of the authors (4.3 to 4.5):
*As suggested, the example shown in section 4.5, that is based on new and unpublished data yet, was removed from the manuscript. The examples of the other sections (4.3, 4.4) were based on previously published data. This fact was added to the legends of the respected figures and tables of the revised manuscript.
This manuscript therefore does not fulfill the requirements of a literature review:
*The present manuscript, that is dealing with a novel TD NMR relaxation sensor for rapidly and accurately monitoring food safety and quality is based on many previously reported studies together with our unique recent published reports about chemical and morphological fingerprinting capacity. We feel that now after improving the references section it is fulfilling the requirements of a review manuscript.
Reviewer 2 Report
The review successfully describes the efficacy of TD NMR sensor for food analysis and identifies previously published research on a topic and summarize the information. But, review article does not report original research of its own, and this has just been done in sections 4.2., 4.3.,4.4. and 4.5. The authors explain their results and compare them with the literature data. It is part of a scientific article but not a review. In my opinion, that part doesn’t go into a review article.
The paper is well written and text is clear, with a large amount of detailed information. Title and abstract actually reflect what is covered in the review. Language and phrasing is clear and unambiguous to avoid confusion.
There are some issues (grammatical errors) that need to be solved and the chapter “References” are not written according to the instructions for authors of the journal (the year of publication must be bold, the title of the journal must be written in abbreviations..).
Author Response
Rev. 2: (a point-by-point response to the reviewer’s comments )
English language and style are fine/minor spell check required:
*As requested, we checked and improved the English and style.
review article does not report original research of its own, and this has just been done in sections 4.2., 4.3.,4.4. and 4.5:
*As requested, we revised these sections (4.2,4.3,4.4, 4.5) showing examples of the TD NMR sensor in several types of food products. We removed section 4.5 that is based on unpublished yet data. The other sections (4.2 – 4.4) are based on previously reported data and we clearly introduced it to the text of the review manuscript.
“References” are not written according to the instructions for authors of the journal (the year of publication must be bold, the title of the journal must be written in abbreviations.):
*As requested, we checked and corrected the references all over the text with emphasis of the references list, according to the instructions of the journal.
Round 2
Reviewer 1 Report
Dear Authors,
Following the explanation of Palmatier et al. (Palmatier, R.W., Houston, M.B. & Hulland, J. Review articles: purpose, process, and structure. J. of the Acad. Mark. Sci. 46, 1–5 (2018). https://doi.org/10.1007/s11747-017-0563-4) review papers “are critical evaluations of material that has already been published,” some that include quantitative effects estimation (i.e., meta-analyses) and some that do not (i.e., systematic reviews). They carefully identify and synthesize relevant literature to evaluate a specific research question, substantive domain, theoretical approach, or methodology and thereby provide readers with a state-of-the-art understanding of the research topic. While I certainty agree with the authors on the wide field of application of TD NMR and that it has many advantages, I still have difficulties to understand the aim of the review article and its purpose. The application of TD NMR fingerprinting is widely reported. There are also many review articles already published.
While the authors now use the commonly understood technical terms in food safety (e.g. it is residues not residuals, it is testing and not assessment (which is a risk management term), they still tend to exaggerated the use of toxic. While we can agree that some risk assessment work comes to the conclusion that PUFAs can have negative effects on human health, to call them “toxic” is simply exaggeration and populistic. A scientific paper should use more scientific care in the use and especially in the repeated use of words like “toxic”.
In food safety related papers the term “consumer” is used, not the term “customer” (line 72).
In food safety related paper the term “processing” is used, not the term “preparation” (line 57).
In food safety related papers the term “food chain” is used, not “food cycle” (line 56).
The authors state in line 61 that current methodologies are not on-line-efficient. There are some methodologies that are not, but there are others that are (FT-IR, Raman, HPLC, etc.).
The authors repeatedly write about “testing of food safety” or “food quality” (for example line 71/72). In general, you cannot test for food safety of food quality. You can test for indicators or parameters that imply or do not imply that food is safe (or unsafe) or a food has a certain quality (also what the term quality in itself needs definition or explanation as quality of food lies in the eye of the beholder).
The sentence from line 81 - 86 is not coherent, it needs to be rephrased.
Lines 106 to 126: In this section the authors do not in detail distinguish between the different analytical techniques. The cost of a FT-IR or Ramen spectrometer is not higher than the cost of a low field NMR. For some application these techniques may also not need sample preparation. The comparison between GC-MS and TD NMR is also done rather poorly, because while a GC-MS might be more expensive, the detection of FAME via GC-MS delivers quantitative results for every single compound. In this case the authors are comparing apples with peas. They should make a contribution to explain the benefits of each techniques for different types of questions and then explain why they believe that for their application TD NMR is more appropriate to use. Simply stating that one instrument is more expensive than the other is not sufficient.
In the section between lines 145 to 151 the authors point out that conventional microbial analytical techniques are time consuming and require specialized and costly equipment. Unfortunately I cannot consent with the authors. Classical microbiological testing requires little instrumentation but only an incubator. It is currently also one the test systems with the lowest costs in food testing. I can consent, that incubation times between 24 and 72 hours might be described as long. But compared to the costs of a low field NMR and the cost of antibody-functionalized polymer-coated magnetic nanoparticles, classical microbiological analysis is still much cheaper and selective in detection.
The authors should use oxygen when they are writing about oxidation processes instead of O2.
In lines 168-172 the authors use “toxic” in relation with microbes. In food safety we usually use words like “non-pathogen” or “pathogen” when microbes are described. They are not described as “toxic” or “not toxic”. TD 1H NMR might be used to monitor compounds from the metabolism of microbes that have a toxicological relevant effect, but is not able to detect microbes directly.
In the sentence from line 208 to 209 the authors use an unsuitable reference. The article by Martinez-Yusta et al. might be a review on TD NMR, but it is not a toxicological review.
While the authors have added 40 references it is still unclear how they conducted the literature search for the review (for example: timeframe of the search, which databases were used for the search, which keyword were used, was grey literature included into the search (and if not, please give a rationale why), etc.). The authors have not addressed my issue about how they made the choice of which references they have chosen and which not (and why).
Author Response
Rev 1: (point-by point response)
Rev 1: “I still have difficulties to understand the aim of the review article and its purpose. The application of TD NMR fingerprinting is widely reported. There are also many review articles already published”
* The aim of our review was introduced at the end of the Introduction section and states the new information presented in this review over what is presently reported:
The original end of the Introduction was “An efficient methodology, is reviewed for overcoming the present analytical limitations for monitoring food’s composition, with an emphasis on oxidized food components such as polyunsaturated fatty acids in complex structures, including food emulsions, using compact instruments for facile real time analysis. This is achieved with an intelligent low field proton NMR as a time domain (TD) NMR relaxation sensor technology monitoring of T2 (spin-spin) and T1 (spin-lattice) energy relaxation times, for testing of food safety and quality, to support decision making of producers, retailers and consumers for food safety and nutritional value during production, shipping, storage and consumption.”
In the revised version, we further added to the above paragraph "Though there are many reports about TD NMR applications as we cite, the ability of using the sensor technology described in this review to produce fingerprints based on chemical and morphological assembly, for food safety analysis is not yet widely known nor available."
Rev 1: “they still tend to exaggerated the use of toxic. While we can agree that some risk assessment work comes to the conclusion that PUFAs can have negative effects on human health, to call them “toxic” is simply exaggeration and populistic. A scientific paper should use more scientific care in the use and especially in the repeated use of words like “toxic”.
* As suggested by the reviewer, we changed the terminology used in the manuscript. More specifically we change "toxic" to "unhealthy" and "unsafe" throughout the manuscript.
Rev 1: “In food safety related papers the term “consumer” is used, not the term “customer” (line 72).”
*As suggested by the reviewer, we changed "customer" to "consumer" throughout the manuscript.
“In food safety related paper the term “processing” is used, not the term “preparation” (line 57).”
*As suggested by the reviewer, we changed "preparation" to "processing" throughout the manuscript.
“In food safety related papers the term “food chain” is used, not “food cycle” (line 56).”
*As suggested by the reviewer, we changed "food cycle" to "food chain" throughout the manuscript.
Rev 1: “The authors state in line 61 that current methodologies are not on-line-efficient. There are some methodologies that are not, but there are others that are (FT-IR, Raman, HPLC, etc.).”
*These mentioned spectral methodologies are very valuable for specific purposes, but they provide one dimensional information regarding specific functional groups and/or separate compounds by different technologies (in the case of HPLC, reverse phase C18, refractometer, etc.). These common methodologies can be used at-line and not on-line, because they require sample extraction from the material being characterized.
Rev 1: “The authors repeatedly write about “testing of food safety” or “food quality” (for example line 71/72). In general, you cannot test for food safety of food quality. You can test for indicators or parameters that imply or do not imply that food is safe (or unsafe) or a food has a certain quality (also what the term quality in itself needs definition or explanation as quality of food lies in the eye of the beholder).”
* The present manuscript is dealing with a sensor for testing a food’s safety and quality. Indeed, we agree with the reviewer that the sensor can test indicators and parameters of safety and quality of foods. Since there are clear definitions of food safety and quality, we feel that we cannot accept this comment of the reviewer, that quality of food lies only in the eye of the beholder. (As an example we can bring the following paragraph cited from the present manuscript "For the detection of unhealthy compounds generated in the oxidation reactions of Scheme 1, it is worth noting, for food safety analysis, LF NMR’s efficacy to detect unhealthy oxygenated α, β-unsaturated aldehydes, like 4-hydroperoxy-, 4,5-epoxy-, and 4-hydroxy-2-alkenals, which are generated in the degradation process of food lipids having omega-3 and omega-6 polyunsaturated groups [19]. These oxidized products are considered as both genotoxic and cytotoxic, and are potential causative agents of cancer, atherosclerosis, and Parkinson’s and Alzheimer’s diseases [6].")
Rev 1: “The sentence from line 81 - 86 is not coherent, it needs to be rephrased.”
* As asked by the reviewer we rephrased this sentence in the manuscript.
Rev 1: “Lines 106 to 126: In this section the authors do not in detail distinguish between the different analytical techniques. The cost of a FT-IR or Ramen spectrometer is not higher than the cost of a low field NMR. For some application these techniques may also not need sample preparation. The comparison between GC-MS and TD NMR is also done rather poorly, because while a GC-MS might be more expensive, the detection of FAME via GC-MS delivers quantitative results for every single compound. In this case the authors are comparing apples with peas. They should make a contribution to explain the benefits of each techniques for different types of questions and then explain why they believe that for their application TD NMR is more appropriate to use. Simply stating that one instrument is more expensive than the other is not sufficient.”
* With all the respect to the reviewer, we think he missed the main point stated in our manuscript that the uniqueness of the described intelligent TD NMR Sensor is its ability to rapidly and accurately follow after the chemical composition and the structural organization of the analyzed food products. In addition to this unique TD NMR sensor technology of material parameters, it is in addition also relatively low cost in comparison to the hardware, software and time needed to generate results for GC-MS.
Rev 1: “In the section between lines 145 to 151 the authors point out that conventional microbial analytical techniques are time consuming and require specialized and costly equipment. Unfortunately I cannot consent with the authors. Classical microbiological testing requires little instrumentation but only an incubator. It is currently also one the test systems with the lowest costs in food testing. I can consent, that incubation times between 24 and 72 hours might be described as long. But compared to the costs of a low field NMR and the cost of antibody-functionalized polymer-coated magnetic nanoparticles, classical microbiological analysis is still much cheaper and selective in detection.”
*Indeed, we agree with the reviewer that the common conservative methods of microbial analytical techniques are simple and very low cost. However, in the 21 century the time required for getting answers is becoming highly important. The Corona pandemic we are presently facing provides an excellent example of the need for a rapid diagnostic system. Time is a very limiting factor and this increases the need of the scientific and technological communities to rapidly develop new diagnosis systems in the coming days as soon as possible. LF H1 NMR offers just such possibilities for rapid material analysis
Rev 1: “The authors should use oxygen when they are writing about oxidation processes instead of O2.”
*As suggested by the reviewer, we changed the text from "O2" to "oxygen" in all places needed.
Rev 1: “In lines 168-172 the authors use “toxic” in relation with microbes. In food safety we usually use words like “non-pathogen” or “pathogen” when microbes are described. They are not described as “toxic” or “not toxic”. TD 1H NMR might be used to monitor compounds from the metabolism of microbes that have a toxicological relevant effect, but is not able to detect microbes directly.”
*We agree with the comment of the reviewer and accordingly changed the text.
Rev 1: “In the sentence from line 208 to 209 the authors use an unsuitable reference. The article by Martinez-Yusta et al. might be a review on TD NMR, but it is not a toxicological review.”
*The reviewer is right that this review reference is focused on TD NMR but it considers relevant review of paper regarding the toxicological aspects. In any case, as requested we addressed this issue in revised text. (we added reference 11 " Bintsis, T. Microbial Pollution and Food Safety. AIMS Microbiol. 2018, 4, 377–396, doi:10.3934/microbiol.2018.3.377.)
Rev 1: “While the authors have added 40 references it is still unclear how they conducted the literature search for the review (for example: timeframe of the search, which databases were used for the search, which keyword were used, was grey literature included into the search (and if not, please give a rationale why), etc.). The authors have not addressed my issue about how they made the choice of which references they have chosen and which not (and why).”
* As requested by the reviewer, using GOOGLE Search with relevant specific key words, we added and improved the references list of the manuscript. The rational that led us was to introduced relevant citations that may support the main topic of the present manuscript dealing with TD NMR sensors for safety and quality of food products. We wished to focus on previous professional publication that are closely related and associated to the core sensor technological message of the present review.
Reviewer 2 Report
The authors accepted most of the suggestions. Grammatical errors and English has been corrected. References have been corrected and written according to the instructions for authors. I think the paper is now suitable for publication
Author Response
Rev 2: (point-by point response)
*Reviewer 2 was satisfied with our first revised version and asked no additional changes.